# Polarization and Incident Angle Independent Multifunctional and Multiband Tunable THz Metasurface Based on VO_2_

**DOI:** 10.3390/nano14121048

**Published:** 2024-06-18

**Authors:** Rehmat Iqbal, Ubaid Ur Rahman Qureshi, Cao Jie, Zia Ur Rahman, Naveed Jafar

**Affiliations:** 1Biomimetic Robots and System, Ministry of Education, School of Optics and Photonics, Beijing Institute of Technology, Beijing 100081, China; rehmat_iqball@outlook.com; 2Beijing Engineering Research Center for Mixed Reality and Advanced Display, School of Optics and Photonics, Beijing Institute of Technology, Beijing 100081, China; 3Yangtze Delta Region Academy, Beijing Institute of Technology, Jiaxing 314003, China; 4State Key Laboratory of Intelligent Control and Decision of Complex System, School of Automation, Beijing Institute of Technology, Beijing 100081, China

**Keywords:** metasurface, terahertz (THz), polarization conversion, absorption, polarization angle, vanadium dioxide (VO_2_)

## Abstract

Aiming at the limitations of single-functionality, limited-applicability, and complex designs prevalent in current metasurfaces, we propose a terahertz multifunctional and multiband tunable metasurface utilizing a VO_2_-metal hybrid structure. This metasurface structure comprises a top VO_2_-metal resonance layer, a middle polyimide dielectric layer, and a gold film reflective layer at the bottom. This metasurface exhibits multifunctionality, operating independently of polarization and incident angle. The varying conductivity states of the VO_2_ layers, enabling the metasurface to achieve various terahertz functionalities, including single-band absorption, broadband THz absorption, and multiband perfect polarization conversion for linear (LP) and circularly polarized (CP) incident waves. Finally, we believe that the functional adaptability of the proposed metasurface expands the repertoire of options available for future terahertz device designs.

## 1. Introduction

Terahertz (THz) technology and its devices have sparked curiosity in researchers’ minds due to the rapid advancement in THz science and technology. Their broad spectrum of applications in non-intrusive testing, spectroscopic detection, security screening, sensing, optical imaging, communications, etc. has intensified the allure and importance of their cutting-edge technology [1]. Traditional THz wave transmission regulation techniques rely mostly on phase accumulation. The immediate interaction of THz radiation with natural materials poses a significant challenge due to the characteristics of THz radiation [2]. Metasurfaces (MSs) have developed as an extremely effective solution to overcome this fundamental constraint. Such metasurfaces are characterized by their exceptionally thin metamaterial structures, comprised of planar electromagnetic (EM) microstructures capable of changing the polarization, amplitude, and phase of electromagnetic waves [3]. Research has explored diverse, linearly polarized (LP) converters [4,5,6,7,8] and circularly polarized (CP) converters [9,10,11,12]. There have also been extensively investigated dual-band [13,14,15], multiband [16,17,18,19], and broadband [20,21,22] absorbers. Therefore, there is a strong emphasis on tunable multifunctional devices exploiting metasurfaces consisting of active functional materials.

Polarization refers to light oscillation, commonly explained through the electric field vector [23,24]. This fundamental property of light finds widespread utility across diverse domains, including quantum optics, imaging, optical displays, light-matter interaction, and sensing [25,26,27,28]. Notably, polarization converters demonstrate the ability to manipulate the polarization of THz waves [29]. Such converters facilitate polarization conversion through several modes, encompassing LP-to-LP [30,31], LP-to-CP [9,32,33,34], and CP-to-LP [35,36] conversions.

Another prominent area of research within the domain of metasurfaces is absorbers, which have emerged as significant contributors in the burgeoning field of energy-manipulating devices, primarily owing to their remarkable light-harvesting capabilities. Capitalizing on their intrinsic high-loss properties, these tailored surfaces offer promising prospects for revolutionizing diverse sectors, including solar energy collection [15], thermal imaging technology [16], and the design of ultra-sensitive photodetectors [17]. Landy et al. pioneered the ideal absorber in 2008, ingeniously merging a split-ring resonator with a metal-wire design to achieve outstanding single-band absorption [18]. Nevertheless, the ability to dynamically modulate the properties of metamaterial is essential for their practical applications. Recognizing this, researchers have begun a material exploration, investigating unconventional platforms such as graphene [20,21,22,37], indium tin oxide (ITO) [23], vanadium dioxide (VO_2_) [24,25], and photosensitive silicon [26,38]. These efforts have resulted in adaptable metasurface absorbers that are not only dual-band but also multiband. Among these, VO_2_ exhibited unique properties of ultrafast stability and conversion rate [39,40]. VO_2_ is a correlated-electron material with an insulator-to-metal phase transition that can be triggered by thermal, electrical, or optical stimuli [41,42]. The transformation of VO_2_ between different states leads to significant differences in their optical and electrical properties.

Previous studies predominantly concentrated on the tunability of a singular functionality, such as single-band absorption [43,44,45]. It also focused on the tunability of broadband polarization conversion for singular functionality in the THz range [46,47,48]. Currently, there is a concerted effort within the scientific community to investigate methodologies for integrating multiple electromagnetic functionalities into a singular device. This pursuit aims to mitigate the manufacturing expenses associated with metasurfaces while broadening their application domains.

So far, limited research endeavors have explored the utilization of metasurfaces to achieve multiband absorption and polarization conversion under linear and circular polarized incident waves while remaining independent of both incident and polarization angles. Moreover, metasurface functional devices face challenges, including low efficiency, restricted bandwidth, lack of tunability, and intricate control methodologies. Hence, our work focuses on designing a multifunctional and multiband metasurface characterized by a simple structure and outstanding performance to facilitate the design of THz photonic devices.

In this paper, we propose a THz multifunctional and multiband tunable metasurface based on a VO_2_-metal hybrid structure. This configuration enables functional switching between single-band and multiband absorption and polarization conversion. Specifically, when the conductivity of VO_2_ is 10 S/m, it behaves as a dielectric, facilitating multiband polarization conversion with a polarization conversion ratio (PCR) exceeding 100% for both LP and CP incident waves at frequencies of 0.11 THz and 0.21 THz, respectively. Upon increasing the conductivity of VO_2_ to 4 × 10^3^ S/m, the metasurface transitions to an absorber, achieving an absorption rate of 86% at 0.11 THz. Additionally, when the conductivity of VO_2_ reaches 2 × 10^5^ S/m, the proposed design attains absorption across an ultra-broadband range spanning from 0.27 THz to 0.36 THz, achieving a rate of 61%.

Moreover, the proposed design exhibits a stable response to incident and polarization angles up to 90°. The distributions of surface-induced currents on the unit cell and the relative impedance are investigated to elucidate the underlying physical mechanisms. This work represents a significant advancement in THz metasurface technology, showcasing potential applications in light manipulation and laying the groundwork for future developments in photonic device integration.

## 2. Structure Design and Simulation

To achieve multiple functionalities from a single device, it is essential to meticulously design the proposed metasurface structure to elicit distinct responses. The proposed metasurface is meticulously crafted from various layers of materials to exhibit multifunctional capabilities. By adjusting the conductivities of VO_2_ material, the topology can be configured to operate as either a multiband polarization converter or multiband absorber for LP and CP incident waves. The schematic representation of the proposed snowflake-shaped metasurface is depicted in Figure 1a; Figure 1b,c show the unit segment’s top view and three-dimensional view.

The optimized geometric measurements of unit cells are as follows: *P* = 300 µm, *G* = 10 µm, *L* = 15 µm, *R* = 145 µm, *D* = 195.06 µm, and *h* = 100 µm, as depicted in Figure 1c. The lossless polyimide, placed between the upper and bottom layers, has a relative permittivity of 3.5 and a tangent loss of 0.02 [49]. Furthermore, the metal surface is comprised of gold (Au) with a conductivity of 4.561 × 10^7^ S/m [50]. The thickness of the VO_2_, gold, and polyimide dielectric layer is reported to be 0.2 μm, 0.2 μm, and 100 μm, respectively. Employing the Drude model, the permittivity of VO_2_ is obtained using Equation (1) [51].
(1)$$\varepsilon \left(\omega \right)=\varepsilon_{\infty }-\frac{\omega_{p}^{2}\left(\sigma \right)}{\left(\omega^{2}+i\gamma \omega \right)}$$
where $\varepsilon_{\infty }=12$, $\gamma ={5.75\times 10}^{13}\;rad/s$, the plasma frequency is obtained at the conductive state (σ) using $\omega_{p}^{2}\left(\sigma \right)=\sigma /\sigma_{0}\omega_{p}^{2}\left(\sigma_{0}\right)$ in which $\sigma_{0}={3\times 10}^{5}\;s/m$ and $\omega_{p}\left(\sigma_{0}\right)=1.4\times 10^{15}\;rad/s$. At $T_{C}=340\;K$, convert the status from insulating to conducting, and the conductivity and permittivity fluctuate dramatically across the insulating to conducting state [52]. In this work, we solely investigate the conductive state of VO_2_ for the polarization conversion and absorption at σ=10 s/m and $\sigma ={2\times 10}^{5}\;s/m$, respectively.

## 3. Metasurface Performance

### 3.1. Performance of the Multifunctional (SFL) Metasurface as a Polarization Conversion

The efficacy of the proposed snowflake (SFL)-shaped metasurfaces can be evaluated through the utilization of a comprehensive full-wave simulation tool, such as CST MW Studio. The unit cell exhibits periodic boundary conditions along the *x* and *y* axes, with an open boundary introduced along the *z* direction to facilitate wave propagation along the *z* axis. This arrangement allows for incident terahertz waves with different polarizations in the frequency range of 0.01 THz to 0.4 THz.

In accordance with the principles of polarization, the Jones matrix can be used to correlate incident polarized waves with their reflected counterparts [34]. The relationship is expressed as in Equation (2).
(2)$$\left(\begin{array}{c} E_{r}^{x}\\ E_{r}^{y} \end{array}\right)=\left(\begin{array}{cc} R_{xx} & R_{xy}\\ R_{yx} & R_{yy} \end{array}\right)\left(\begin{array}{c} E_{i}^{x}\\ E_{i}^{y} \end{array}\right)=R\left(\begin{array}{c} E_{i}^{x}\\ E_{i}^{Y} \end{array}\right)$$

Herein, *R* signifies the Cartesian Jones reflection matrix. It operates in collaboration with the incident electric field $E_{i}^{x\left(y\right)}$ and reflected electric field $E_{r}^{x\left(y\right)}$, aligned in $x\left(y\right)$ directions. The reflection matrix for circularly polarizations is attained by employing the linear polarized reflection coefficient subsequent to the conversion from Cartesian to circular base.
(3)$$R_{CP}=\left(\begin{array}{cc} R_{++} & R_{\pm }\\ R_{\mp } & R_{--} \end{array}\right)=\wedge^{-1}R\wedge \;=\;\frac{1}{2}\left(\begin{array}{cc} R_{xx}-R_{yy}-i\left(R_{xy}+R_{yx}\right) & R_{xx}+R_{yy}+i\left(R_{xy}-R_{yx}\right)\\ R_{xx}+R_{yy}-i\left(R_{xy}-R_{yx}\right) & R_{xx}-R_{yy}+i\left(R_{xy}+R_{yx}\right) \end{array}\right)$$

Herein, ∧ denotes the coordinate transformation matrix is utilized for conversion from Cartesian to circular base, defined as $\wedge =\frac{1}{\sqrt{2}}\left(\begin{array}{cc} 1 & 1\\ i & -i \end{array}\right)$. Here, in Equation (3), the signs ‘+’ and ‘−’ indicate the right-handed and left-handed circularly polarized waves, respectively. The unit cell reflection coefficients for normal *x* and *y* incidences or RCP and LCP polarized waves are depicted in Figure 2, where $\left(\frac{R_{xx}}{R_{yy}}\right)$, $(\frac{R_{++}}{R_{--}})\;and\;(\frac{R_{yx}}{R_{xy}})$, $\left(\frac{R_{-+}}{R_{+-}}\right)$ represent the reflection coefficients for co and cross-polarized waves, respectively.

When the conductivity of VO_2_ is 10 S/m, the proposed structure acts as a multiband polarization conversion for both LP and CP incident waves. In Figure 2a,b, when the incident wave is LP and CP, the cross-polarized reflection coefficient ($R_{yx}$, $R_{xy}$, $R_{-+}$, $R_{+-}$) is notably observed to attain a value of 100% at 0.11 THz and 0.26 THz, also achieved more than 50% values in bandwidth (0.10 THz–0.15 THz) and (0.22 THz–0.30 THz), while co-polarization reflection coefficient reaches ($R_{xx}$, $R_{yy}$, $R_{++}$, $R_{--}$) to 0.1 at frequency 0.11 THz and 0 at 0.26 THz. Hence, the envisioned metasurface configuration adeptly transforms linear and circular polarizations into their corresponding cross-polarizations across multiple bands. Despite the absence of C4 symmetry within the structure configuration, a salient feature lies in its inherent and displays mirror symmetry along the *u*-axis, resulting in the equivalence of $R_{xx}=R_{yy}$, $R_{yx}=R_{xy}$, $R_{-+}=R_{+-}$ and $R_{++}=R_{--}$.

The analysis of the proposed metasurface, cross-polarized conversion (CPC), is further elucidated through the calculation of the polarization conversion ratio (*PC*R), as defined by Equation (4).
(4)$$PCR=\frac{{\left|R_{cross}\right|}^{2}}{{\left|R_{cross}\right|}^{2}+{\left|R_{co}\right|}^{2}}$$

Figure 2c,d illustrate the *PC*R associated with LP and CP, respectively. Notably, within the frequency range of (0.1 THz–0.4 THz), the *PC*R attains 100% efficiency within specific frequencies, namely 0.11 THz and 0.26 THz, and more than 90% at frequency intervals (0.1 THz–0.12 THz) and (0.261 THz–0.276 THz). The polarization conversion ratio is the same for both LP and CP incident waves due to the unique structure design of the unit cell.

In practical applications, the assessment of metasurface performance often necessitates consideration of wide-angle incidence scenarios. Figure 3 depicts the influence of incident LP and CP light at various angles of incidence and azimuth on the polarization conversion effect within metasurface structures. Figure 3a illustrates the consistent polarization conversion ratio (PCR) within an incidence angle range of 0° to 85°. Additionally, Figure 3b provides an examination of diverse azimuthal incidences on PCR, indicating a stable PCR bandwidth within an azimuthal angle range of 0° to 85°. This observed angular stability is ascribed to the diminutive dielectric thickness and unit cell size. Considering the potential for incoming waves to exhibit arbitrary incidence angles in practical scenarios, the metasurface’s insensitivity to azimuth and incidence angles renders it a promising candidate for a variety of applications [53].

Furthermore, the physical mechanism is essential to analyzing the performance of the polarization conversion of the proposed metasurface. Therefore, considering the surface distribution at the different frequencies, namely 0.106 THz, 0.118 THz, 0.14 THz, 0.244 THz, 0.268 THz, and 0.29 THz, we have the top gold surface of the proposed SFLM structure and the bottom layer (ground plan) of the unit cell for *y*-polarized incident waves. According to Faraday’s law, a changing magnetic field between the two metals causes surface current to flow in opposite directions on the top and bottom of the metallic layers. The black-colored arrow indicates the net current shown in Figure 4. Equation (5) can be used to determine the relationship between electric and magnetic dipole moments and average electromagnetic fields.
(5)$$\left[\begin{array}{c} p\\ m \end{array}\right]=\left[\begin{array}{cc} \alpha_{ee} & \alpha_{em}\\ \alpha_{me} & \alpha_{mm} \end{array}\right]\left[\begin{array}{c} E\\ H \end{array}\right]$$

Here, P = ${\left[\begin{array}{cc} p_{x} & p_{y} \end{array}\right]}^{T}$, $m={\left[\begin{array}{cc} m_{x} & m_{y} \end{array}\right]}^{T}$ is the electric dipole moment and magnetic dipole moment, respectively, and the matrix (α) contains the materials coefficients in terms of electric and magnetic field, while the $E={\left[\begin{array}{cc} E_{x} & E_{y} \end{array}\right]}^{T}$ and $H={\left[\begin{array}{cc} H_{x} & H_{y} \end{array}\right]}^{T}$ is the average tangential electric and magnetic fields, respectively, at the metasurface. The time-changing electric and magnetic surface current polarization will cause electric and magnetic surface currents on the metasurface, which is expressed in Equation (6):(6)$$\left[\begin{array}{c} J\\ M \end{array}\right]=i\omega \left[\begin{array}{cc} \alpha_{ee} & \alpha_{em}\\ \alpha_{me} & \alpha_{mm} \end{array}\right]\left[\begin{array}{c} E\\ H \end{array}\right]$$
where $J={\left[\begin{array}{cc} J_{x} & J_{y} \end{array}\right]}^{T}$ and $M={\left[\begin{array}{cc} M_{x} & M_{y} \end{array}\right]}^{T}$ electric and magnetic current densities, respectively, and ω is the angular frequency of the incident electromagnetic wave. The relationships between the surface current density *J* and radiated for fields are given by
(7)$$E=-i\frac{\omega \mu }{4\pi }\int J\left(x,y\right)\frac{e^{-ikR}}{R}dxdy$$

According to Equation (7), for CPC, an electric field that is polarized along the *x* direction will result in current flow on the metasurfaces in the *y* direction, while an electric field polarized along the *y* direction will induce current flow in the *x* direction. Figure 4a–c,g–i demonstrates the cross-polarization characteristics at various operating frequencies. At 0.10 THz, 0.11 THz, and 0.14 THz, the surface current on the top metasurface and the ground plate is inversely parallel, leading to the excitation of magnetic resonance and the generation of an induced magnetic field. This induced magnetic field results in the cross-polarization effect where the reflected THz wave becomes *x*-polarized, as indicated by Equation (6). Similarly, Figure 4d,j illustrate that, at a frequency of 0.24 THz, the surface current on the top layer becomes corresponding to the ground layer, leading to the generation of electric resonance and the formation of an induced electric field. By decomposing the electric field into its orthogonal components *x* and *y*, it is evident that the electric field along the *x* axis can cross-couple with the incident electric field to form CPC and result in the reflected THz wave being *x*-polarized according to Equation (5). Figure 4e,f,k,l illustrate that, at frequencies 0.26 THz and 0.29 THz, the primarily distributed surface current on the top layer results in the creation of electrical resonance, forming an electric dipole, where the *x* component of the induced electric field plays a significant role in generating cross-polarization effect. These observations further validate the cross-polarization characteristics and support the electromagnetic behavior described by the equation, showcasing the potential applications of these findings in THz wave manipulation and control.

Furthermore, the phenomena of cross-polarization conversion can be comprehended through the inherent anisotropy in the designed structure. By rotating the standard *x*-*y*, the coordinate system is rotated 45° to establish a unique *u*-*v* coordinate system. As shown in Figure 5a, the anisotropic nature of the snowflake-like structure allows the incoming *y*-polarized wave to be divided orthogonally into constituents along the *u* and *v* axes. In resulting of this division enables the mathematical representation of the incident and reflected waves as follows [46]:$\vec{E_{i}}=\left(\vec{u}E_{ui}+\vec{v}E_{vi}\right)exp(i\varphi )$ and $\vec{E_{r}}=\left(\vec{u}{R_{uu}E}_{ui}+\vec{v}{E_{vv}E}_{vi}\right)exp(i\varphi )$. Here, $R_{uu}=\left|E_{ur}\right|/\left|E_{ui}\right|$ and $R_{vv}=\left|E_{vr}\right|/\left|E_{vi}\right|$ represent the reflection coefficients in the *u* and *v* directions, respectively, with subscripts *i* and *r* indicating incident and reflected waves.

Moreover, the phase disparity (Δφ) arises from structural asymmetry, leading to the relationship between reflection coefficients can be written as: $R_{vv}=R_{uu}\exp(j\Delta \varphi ).$ Figure 5b depicts the reflection coefficients of the reflected wave, while Figure 5c depicts the phase and phase differences. In the frequency range of 0.15 to 0.4 THz, the reflection coefficients $R_{uu}$ and $R_{vv}$ approaches to 1, with a phase difference of around ±180°. These results indicate that the synthesized waves of $R_{uu}$ and $R_{vv}$ deviate by 90° from the incident wave, showcasing the metasurface’s capability to transform *y*-polarized incident waves into *x*-polarized reflected waves.

### 3.2. Performance of the Multifunctional (SFL) Metasurface as an Absorption

As well as the performance of the proposed metasurface in polarization conversion, the multifunctional (SFL) metasurface was designed to operate as an absorber owing to the inherent conductivity of vanadium dioxide (VO_2_). As depicted in Figure 1, a strategically placed VO_2_ film on the corner of the snowflake-like structure is instrumental in constituting the multiband absorber. 

The absorptivity of the multifunctional (SFL) metasurface is determined through the application of Equation (6).
(8)A(w)=1−R(w)−T(w)

Herein, $R={\left|S_{11}\right|}^{2}$ represent the reflectivity, $T={\left|S_{21}\right|}^{2}$ signifies the transmissivity of the metasurface. It is noteworthy that the gold is utilized as the reflector, resulting in the transmission being zero, *T* = 0. The absorptivity of the metasurface for LP waves is computed as an equation
(9)$$A(\omega )=1-{\left|R_{xx}\right|}^{2}-{\left|R_{yx}\right|}^{2}$$

For CP wave,
(10)A(ω)=1−R++−−2−R−+/+−2

According to Equations (7) and (8), when the incident wave is composed of both LP and CP components and given a conductivity of VO_2_ as 4 × 10^3^ S/m, an absorption rate of 86% is attained at 0.11 THz. Moreover, an absorption exceeding 50% is achieved within the frequency bandwidth ranging from 0.09 THz to 0.15 THz, as illustrated in Figure 6a,b.

When the conductivity of VO_2_ is increased to 2 × 10^5^ S/m, the absorption of 61% is obtained at 0.34 THz and achieved more than 50% absorption in bandwidth (0.27 THz–0.36 THz) for LP and CP incident wave, as shown in Figure 6c,d.

Impedance-matching theory is an essential principle for evaluating absorption. In this study, the absorption and relative impedance is achieved [28].
(11)$$A=1-R=1-{\left|\frac{{Z-Z}_{air}}{{Z-Z}_{air}}\right|}^{2}=1-{\left|\frac{Z_{r}-1}{Z_{r}+1}\right|}^{2}$$
(12)$$Z_{r}=\sqrt{\frac{{\left(1+S_{11}\right)}^{2}-S_{21}^{2}}{{\left(1-S_{11}\right)}^{2}-S_{21}^{2}}}$$
where $Z_{air}$ is the impedance of the free space and *Z* is the impedance of the metasurface. The absorption of the metasurface occurs when the effective impedance of the multiband absorber aligns with that of the free space. As depicted in Figure 7a,b, the real part approaches unity, and the imaginary part approaches zero, which is calculated by employing Equations (8) and (9) for the conductivity of the VO_2_ is 4 × 10^3^ S/m and 2 × 10^5^ S/m, respectively.

The necessity for tunability and angular stability is paramount across numerous prospective applications. To ascertain the tunability, an examination of absorption across various conductivities of VO_2_ is conducted, with the findings consolidated in Figure 8. As the conductivity of the VO_2_ increases from 10 S/m to 2 × 10^5^ S/m, there is a corresponding rise in plasmonic absorption within the VO_2_ layer, resulting in an enhancement in the absorption of incident light. However, at extremely high conductivities, plasmonic absorption declines due to the heavy damping of plasmons, weakening their coupling with incident light and decreasing the absorption rate. At 0.11 THz, absorption reaches to 86% with the conductivity of VO_2_ is 4 × 10^3^ S/m and 61% with the conductivity of VO_2_ is 2 × 10^5^ S/m (conductive state), dropping to 0% at 10 S/m (insulating state), demonstrating adjustable multiband absorption from 0% to 86%, as shown in Figure 8a.

Furthermore, the impact of oblique incident angles is examined when the conductivity of VO_2_ is 4 × 10^3^ S/m and 2 × 10^5^ S/m, respectively, with the result shown in Figure 8b,c. The investigation reveals the proposed structure robustness resilience across a wide incidence angle spectrum (0° to 85°). Meanwhile, the absorption performance is investigated for LP and CP incident waves at the conductivity of VO_2_ is 4 × 10^3^ S/m and 2 × 10^5^ S/m, respectively, which demonstrates polarization independence when the incident terahertz wave is normally incident in wide incidence angles spectrum (0° to 85°), owing to the unit cell’s unique rotational symmetry structure, as shown in Figure 8d,e. The stability of the large incident angles and wide polarization angles stems from the unique configuration of the unit cell [54]. As a result, the proposed SFL metasurface design for the absorber has excellent characteristics of wide-angle incidences and wide polarization incidences, which are significant for practical applications.

## 4. Potential Fabrication Process of Designed Metasurface

In the fabrication of the designed metasurface, a combination of sputtering deposition technology [55] and lithography technology [56] is employed. (a) Initially, a layer of polyimide dielectric is deposited onto a silicon wafer via a spin-coating process; (b) Subsequently, a thick layer of gold film is deposited onto the sufficiently thick polyimide substrate utilizing electron beam evaporation; (c) The gold antenna pattern is then formed through lithography and metallization techniques; (d) Following this, a pre-prepared VO_2_ colloid is spin-coated into the gaps of the gold antenna pattern layer, resulting in the formation of a VO_2_ film with the required thickness; (e) The VO_2_ patches structure is further generated using lithography technology; (f) Finally, another deposition of a thick layer of gold film is applied to the back side of the polyimide substrate via electron beam evaporation. This multi-step process ensures the precise fabrication of the metasurface structure with the desired properties.

## 5. Conclusions

In summary, this research article represents the multiple functionalities of a THz metasurface based on vanadium dioxide (VO_2_), which has the functionalities of multiband absorption and multiband polarization conversion. When VO_2_ is an insulating phase, the designed SFLM acts as a multiband polarization converter within the frequencies range of 0.11 THz–0.12 THz and 0.26 THz–0.27 THz; the designed metasurface exhibits robust capability for converting linear and circular polarizations to their respective cross-polarizations When the VO_2_ transitions into a conducting state, the proposed metasurface operates as a multiband absorber, demonstrating absorption peaks at three distinct frequencies. Specifically, absorption rates exceeding 86% at 0.11 THz, over 60% at 0.288 THz, and 61% at 0.343 THz are attainable. Notably, the investigation emphasizes the stability of polarization conversion and absorption spectrum across varying polarization angles while sustaining remarkable multiband absorption capabilities even under high incidence angles. This multifunctional metasurface thus presents promising avenues for advancing future terahertz (THz) devices.

## Figures and Tables

**Figure 1 nanomaterials-14-01048-f001:**
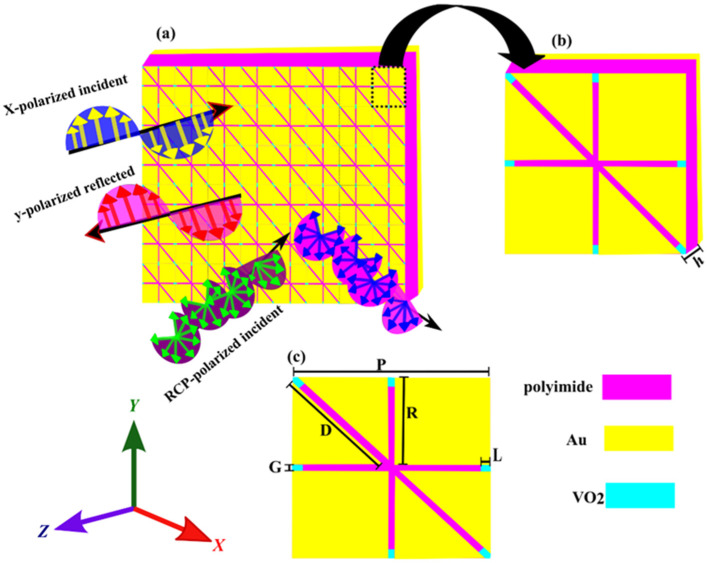
Schematic representation of the multifunctional tunable metasurface (**a**) A two-dimensional arrangement of the constituent unit cells, (**b**) A three-dimensional depiction of the envisaged metasurface, (**c**) Top-down representation of the unit cell.

**Figure 2 nanomaterials-14-01048-f002:**
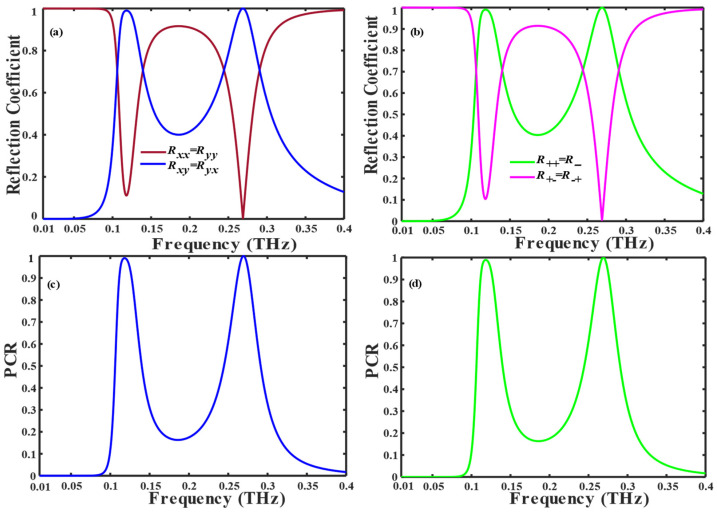
In insulating state VO_2_ = 10 S/m: co- and cross-polarized reflection coefficient for (**a**) LP and (**b**) CP incident waves. Polarization conversion ratio (PCR) of the (**c**) LP and (**d**) CP incident waves.

**Figure 3 nanomaterials-14-01048-f003:**
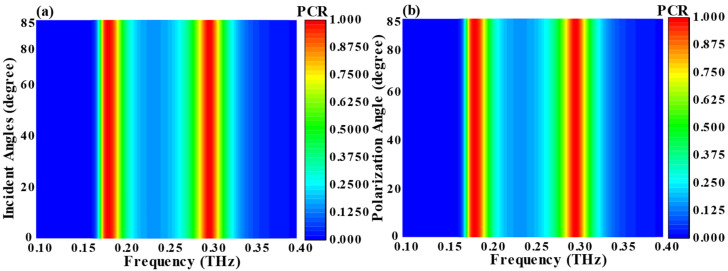
(**a**) Impact of incident angle and (**b**) impact of polarization angle on the PCR.

**Figure 4 nanomaterials-14-01048-f004:**
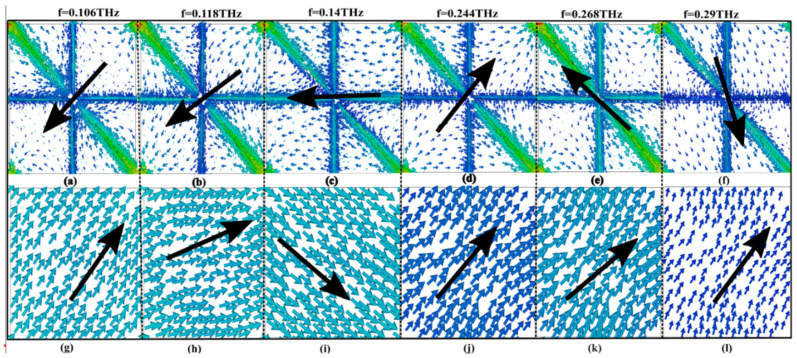
Simulated distributions of the surface current: (**a**) and (**g**) at f = 0.106 THz, (**b**) and (**h**) at f = 0.118 THz, (**c**) and (**i**) at f = 0.140 THz, (**d**) and (**j)** at f = 0.268 THz, (**e**) and (**k**) at f = 0.244 THz, (**f**) and (**l**) at f = 0.290 THz.

**Figure 5 nanomaterials-14-01048-f005:**
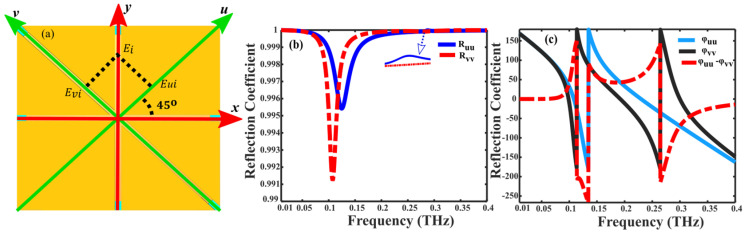
(**a**) Principle of polarization Conversion Operation (**b**) reflection coefficient and (**c**) reflection phases under the *u* and *v*-polarized incident waves.

**Figure 6 nanomaterials-14-01048-f006:**
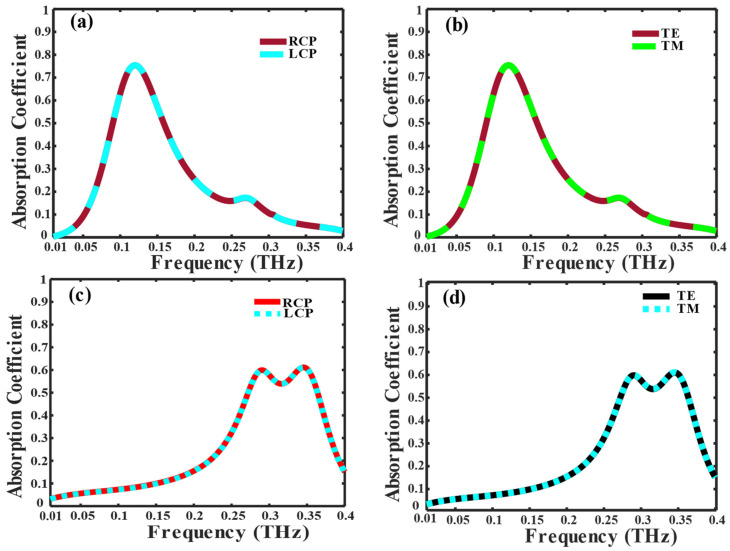
The absorption coefficient of different VO_2_ conductivities: (**a**) for the incident RCP and LCP wave, (**b**) for TE and TM incident waves, at 4 × 10^3^ S/m, respectively, (**c**) for the incident wave RCP and LCP wave, (**d**) for the TE and TM incident wave at 2 × 10^5^ S/m, respectively.

**Figure 7 nanomaterials-14-01048-f007:**
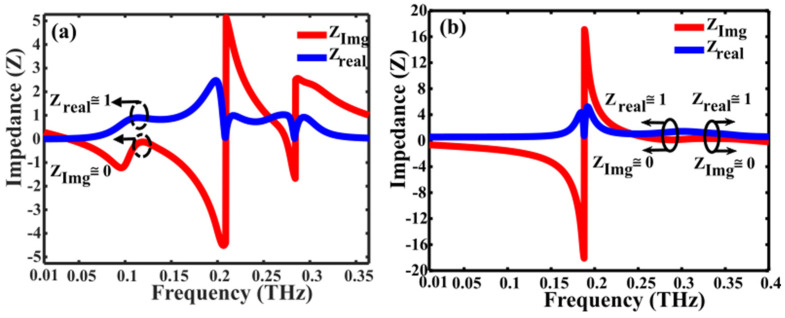
Relative impedance of the multiband absorber with VO_2_ conductivities, 4 × 10^3^ S/m and 2 × 10^5^ S/m at (**a**) f = 0.11 THz (**b**) f = 0.28 THz and 0.34 THz, respectively.

**Figure 8 nanomaterials-14-01048-f008:**
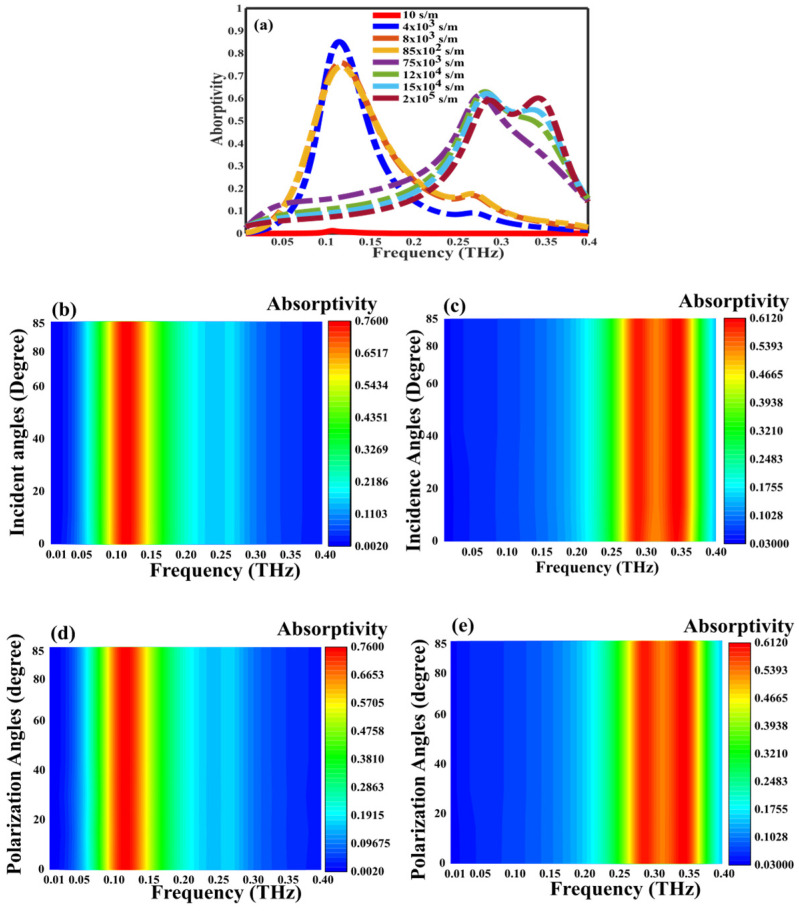
Influences of conductivity, incident angles and polarization angles on absorptivity at different VO_2_ conductivity, 4 × 10^3^ S/m and 2 × 10^5^ S/m: (**a**) different conductivities. (**b**,**c**) influences of incident angles. (**d**,**e**) influences of polarization angles.

## Data Availability

Data will be made available on request.

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
