# Peer review of "Polarization and Incident Angle Independent Multifunctional and Multiband Tunable THz Metasurface Based on VO2"

_nanomaterials, 2024, doi:10.3390/nano14121048_

Round 1
Reviewer 1 Report
Comments and Suggestions for Authors
This paper shows the calculated results of using V0S2 metasurfaces that can provide multi-functions in tunable THz frequency. While this paper provides useful results that will important applications if the computation design proposed by the authors can be fabricated experimentally. I would recommend for publication after the following revisions:
1. In reading the paper, it gives a wrong impression that VO2 materials are new unexplored materials, so it is important to introduce VO2 materials and its applications in photonic, optics and engineering applications. I suggest some papers listed below from 5 different journals:
https://www.degruyter.com/document/doi/10.1515/nanoph-2012-0028/html
https://iopscience.iop.org/article/10.1088/1361-6528/aa9cb1
https://pubs.acs.org/doi/abs/10.1021/acsanm.2c0053
https://www.sciencedirect.com/science/article/abs/pii/S003040182100691X
https://www.sciencedirect.com/science/article/abs/pii/S092596352400075X
2. It is a bit weird to me if this simple 3-layers structures has never been fabricated, I would like the authors to do a literature to report any past studies to comment if the proposed design in this paper can be realized within the current technologies. They should explain the novelty of their works compared to these works.
3. There are many angle-insensitive and broadband tunable meta-surfaces operated at lower frequency [see below]. May be the authors can comment why it is difficult to extend the same old concepts and materials from low frequency and THz and VOS2 is one of the suitable materials to do so.
https://ieeexplore.ieee.org/document/9238433
https://ieeexplore.ieee.org/abstract/document/10027866
Comments on the Quality of English Language
Strongly to suggest having a colleague who is good in English to correct the errors.
Author Response
Reviewer: 1
This paper shows the calculated results of using VO2 metasurfaces that can provide multi-functions in tunable THz frequency. While this paper provides useful results that will important applications if the computation design proposed by the authors can be fabricated experimentally. I would recommend for publication after the following revisions.
We sincerely thank you for your constructive feedback and for recognizing the potential applications of our work on VO2 metasurfaces in tunable THz frequency range. Your comments have highlighted important aspects of our research and have provided us with a clear direction for revisions. We are diligently working to address the revisions you have recommended to ensure that our paper meets the high standards of publication. Your endorsement for publication upon completion of the suggested revisions is greatly appreciated, and we are committed to making the necessary modifications as quickly as possible.
- Comment-1:
In reading the paper, it gives a wrong impression that VO2 materials are new unexplored materials, so it is important to introduce VO2 materials and its applications in photonic, optics and engineering applications. I suggest some papers listed below from 5 different journals:
https://www.degruyter.com/document/doi/10.1515/nanoph-2012-0028/html
https://iopscience.iop.org/article/10.1088/1361-6528/aa9cb1
https://pubs.acs.org/doi/abs/10.1021/acsanm.2c0053
https://www.sciencedirect.com/science/article/abs/pii/S003040182100691X
https://www.sciencedirect.com/science/article/abs/pii/S092596352400075X
Answer:
We appreciate your valuable suggestions, and we have taken them into consideration. The mentioned research articles have been thoroughly reviewed and discussed in the introduction section of the revised manuscript. They have also been included as reference [39,40,41,42].
- Comment-2:
It is a bit weird to me if this simple 3-layers structures has never been fabricated, I would like the authors to do a literature to report any past studies to comment if the proposed design in this paper can be realized within the current technologies. They should explain the novelty of their works compared to these works.
Answer:
Thank you for your valuable feedback. After conducting a thorough literature review, we have added a new section titled "Potential Fabrication Process of Designed Metasurface" in the revised manuscript. In this section, we detail the fabrication process using current technologies and discuss the feasibility of realizing the proposed design. Additionally, we have elaborated on the novelty of our work by comparing it with past studies in the introduction section, highlighting the unique aspects and advancements introduced by our approach.
We included the following discussion in the revised manuscript.
Potential fabrication process of designed metasurface:
In the fabrication of the designed metasurface, a combination of sputtering deposition technology [R-55] and lithography technology [R-56] is employed. (a) Initially, a layer of polyimide dielectric is deposited onto a silicon wafer via a spin coating process ; (b) Subsequently, a thick layer of gold film is deposited onto the sufficiently thick polyimide substrate utilizing electron beam evaporation; (c) The gold antenna pattern is then formed through lithography and metallization techniques ; (d) Following this, a pre-prepared VO2 colloid is spin-coated into the gaps of the gold antenna pattern layer, resulting in the formation of a VO2 film with the required thickness; (e) The VO2 patches structure is further generated using lithography technology ; (f) Finally, another deposition of a thick layer of gold film is applied to the back side of the polyimide substrate via electron beam evaporation . This multi-step process ensures the precise fabrication of the metasurface structure with the desired properties.
- Comment-3:
There are many angle-insensitive and broadband tunable meta-surfaces operated at lower frequency [see below]. May be the authors can comment why it is difficult to extend the same old concepts and materials from low frequency and THz and VO2 is one of the suitable materials to do so.
https://ieeexplore.ieee.org/document/9238433
https://ieeexplore.ieee.org/abstract/document/10027866
Answer:
Thank you for your insightful comments. We appreciate the references to the published work on angle-insensitive and broadband tunable metasurfaces operating at lower frequencies.
In response, we have thoroughly reviewed the mentioned research articles and included a detailed discussion in the revised manuscript. Here, we explain why it is challenging to extend the same concepts and materials from lower frequencies to the THz range and highlight the suitability of VO2 for this application.
The mentioned published works operate at lower frequencies where tuning mechanisms, such as varactor and PIN diodes, are effective. However, these diodes struggle at the much higher frequencies of the THz range due to limitations in their physical properties and response times. At THz frequencies, the efficiency of diodes decreases significantly due to increased parasitic effects and reduced quality factors. Additionally, power dissipation in diodes can become substantial at these higher frequencies, leading to overheating and potential damage.
In the THz range, alternative technologies such as phase change materials (PCMs), graphene, ITO, and VO2 offer more practical and effective solutions for tunability. These materials can exhibit significant changes in their electromagnetic properties when subjected to external stimuli, making them more suitable for THz applications. Among these materials, VO2 stands out due to its unique properties, including ultrafast stability and conversion rates. VO2 is a correlated-electron material that undergoes an insulator-to-metal phase transition triggered by thermal, electrical, or optical stimuli. This phase transition results in significant differences in the material’s optical and electrical properties, making VO2 an excellent candidate for tunable metasurfaces in the THz range.
We have incorporated these points into the revised manuscript to provide a comprehensive comparison and to highlight the novelty and feasibility of our proposed design using current technologies.
We thank Reviewer 1 for the important and helpful comments which has significantly improved the quality of our work.
Reviewer 2 Report
Comments and Suggestions for Authors
The authors numerically proposed a tunable metasurface with polarization and incident angle independent features in their manuscript. The tunability is based on the metal-insulator phase transition in VO2 which is used for a part of metasurface. The design and functionality of the metasurface are interesting. However, the manuscript has a room for improvement in the presentation. I strongly recommend the authors check their manuscript before the submission.
Questions and Comments
1. As for the metasurface structure, the authors should describe the information about the thickness of each layer of polyimide, Au, and VO2. Furthermore, a sideview of the metasurface to depict the layers can help the readers to understand the proposed situation.
2. In the middle part of Eq.(2), the x-component of the E-field was Eiy. But I guess it is Eix. Is it ture?
3. In line 161, the author used “u-axis” without any explanation. They should define the new axis. I recommend the u- and v-axes are defined in Figure 5(a).
4. In line 178, the authors mentioned as “Figure 3 (b) provides an examination of diverse azimuthal incidences on PCR, indicating a stable PCR bandwidth within an azimuthal angle range of 0° to 90°.” However, the maximum value of the vertical axis in Figure 3(b) was 85°. The authors should check, again.
5. In Eq. (5), there is no definition about \alpha_**. The authors should describe the explanation about \alpha.
6. In line 211, the authors mentioned as “At 1.1 THz, 0.11 THz, and 0.14 THz, the surface current on the top metasurface and the ground plate is inversely parallel, leading to the excitation of magnetic resonance and the generation of an induced magnetic field.” However, there is no figure for 1.1 THz in Figure 4. The authors should check the figure.
7. In figure 2 and Figure 5(c), there are a spectral structure in 0.25 THz. However, there is no structure at 0.25 THz in Figure 5(b). The author should discuss the reason why the spectral structure at 0.25 THz is disappeared in the reflection of uv-coordinates.
8. In the caption of Figure 7, the authors should describe the information about the conductivity of VO2.
Author Response
Reviewer: 2
The authors numerically proposed a tunable metasurface with polarization and incident angle independent features in their manuscript. The tunability is based on the metal-insulator phase transition in VO2 which is used for a part of metasurface. The design and functionality of the metasurface are interesting. However, the manuscript has a room for improvement in the presentation. I strongly recommend the authors check their manuscript before the submission.
Answer:
We greatly appreciate your recognition of the novel aspects of our proposed tunable metasurface, as well as the suggestion to improve the manuscript's presentation. Our design aims to explore the potential of VO2 in achieving polarization and incident angle independence, also the transition of VO2 from metal to insulator phase is used for multifunctionalities of metasurface, which we believe presents a valuable contribution to the field. We have carefully reviewed the manuscript to enhance clarity and presentation, and we trust that these improvements will meet your expectations. Thank you for your constructive feedback, which has been helpful in refining our work.
- Comment-1:
As for the metasurface structure, the authors should describe the information about the thickness of each layer of polyimide, Au, and VO2. Furthermore, a sideview of the metasurface to depict the layers can help the readers to understand the proposed situation.
Thank you for your inquiry regarding the thicknesses of the polyimide, Au and VO2 layer. We appreciate your attention to detail. In the revised manuscript, we have included the following thickness values based on our simulation:
- Polyimide thickness: 100 µm
- Au thickness: 0.2 µm
- VO2 thickness: 0.2 µm
- Comment-2:
In the middle part of Eq.(2), the x-component of the E-field was . But I guess it is . Is it ture?
Thank you for pointing out the potential discrepancy in Eq (2). Upon revisiting the equation and the context in which it is presented, we concur with your observation that the correct term should indeed be for the x-component of the electric field. The manuscript has been amended to reflect this correction. We appreciate your attention to detail, which has contributed to the accuracy of our study.
- Comment-3:
In line 161, the author used “u-axis” without any explanation. They should define the new axis. I recommend the u- and v-axes are defined in Figure 5(a).
We appreciate your careful reading of our manuscript and the constructive comment regarding the introduction of the 'u-axis' in line 161. We acknowledge the oversight and agree that a clear definition of the u-v axis is crucial for comprehending our findings. In light of this, we have amended Figure 5(a) to include definitions of both the 'u-' and 'v-axes,' ensuring that their orientation and relevance to the study are clearly conveyed to the reader. This change in the revised manuscript will enhance the clarity and the subsequent results discussed. Once again thank you for your valuable contribution to improving the quality of our work.
- Comment-4:
In line 178, the authors mentioned as “Figure 3 (b) provides an examination of diverse azimuthal incidences on PCR, indicating a stable PCR bandwidth within an azimuthal angle range of 0° to 90°.” However, the maximum value of the vertical axis in Figure 3(b) was 85°. The authors should check, again.
Thank you for your meticulous review and for bringing this discrepancy to our attention. We have re-examined the mentioned section in line 178 along with Figure 3(b) and acknowledge the error in reporting the azimuthal angle range. We have corrected the manuscript to accurately reflect the data presented in Figure 3(b) and the range mentioned in the description, ensuring that the values and descriptions are consistent and precise. We are committed to maintaining the highest level of accuracy in our work and are grateful for your assistance in this matter.
- Comment-5:
In Eq. (5), there is no definition about \alpha_**. The authors should describe the explanation about \alpha.
We are grateful for your valuable recommendation regarding the explication of the alpha (α) Eq. (5). The matrix (α) contains the response coefficients terms of electric and magnetic field. (αe) and (αm) represent the electric and magnetic polarizabilities, respond to the incident electric and magnetic field, respectively. In response to your valuable feedback, we have revised the manuscript to incorporate a precise definition of (α). This amendment will ensure clarity and facilitate a better understanding of the equation.
- Comment-6:
In line 211, the authors mentioned as “At 1.1 THz, 0.11 THz, and 0.14 THz, the surface current on the top metasurface and the ground plate is inversely parallel, leading to the excitation of magnetic resonance and the generation of an induced magnetic field.” However, there is no figure for 1.1 THz in Figure 4. The authors should check the figure.
We appreciate your follow-up and the clarification regarding the frequency of interest. The proposed design functions within the frequency range of 0.1 THz to 0.4 THz. Having reexamined the relevant section in line 211 and Figure 4, we have made the necessary modification to accurately reflect the correct frequency of 0.1 THz. We apologize for any confusion arising from this mistake and are thankful for the opportunity to ensure the precision of our manuscript. The text now correctly cites 0.1 THz, aligning with the presented data in Figure 4.
- Comment-7:
In figure 2 and Figure 5(c), there are a spectral structure in 0.25 THz. However, there is no structure at 0.25 THz in Figure 5(b). The author should discuss the reason why the spectral structure at 0.25 THz is disappeared in the reflection of u-v-coordinates.
I would like to clarify the distinctions between the figures and their implications for the spectral structure observed at 0.25 THz. Figure 2 illustrates the reflection coefficient for a normal incident wave under both linear and circular polarization. In contrast, Figures 5(b) and 5(c) depict the reflection characteristics when the u- and v- axes are oriented at ±45° to the x- and y-axes, respectively. These orientations are employed to elucidate the physical mechanism behind polarization conversion. To elaborate, within a specific frequency range, if ≈ ≈1 and one component of the incident wave is reflected in-phase (phase difference ∆φ ≈ 0°) while the other component is reflected out-of-phase (phase difference ), the electric field of the reflected wave is rotated by 90° relative to the electric field of the incident wave, as shown in Figure 2. This behavior accounts for the spectral structure at 0.25 THz.
The absence of a clear spectral structure at 0.25 THz in Figure 5(b) arises from the specific orientation and scaling used in this figure. To address this, we have amended the graph to include a zoomed-in view of the spectral region around 0.25 THz. This enhancement highlights the subtle spectral structure that was not previously evident due to the original scale of the graph.
- Comment-8:
In the caption of Figure 7, the authors should describe the information about the conductivity of VO2.
We appreciate your valuable suggestion to include details about the conductivity of VO2 in the caption of Figure 7. The oversight has been rectified and the caption has now been updated to accurately describe the VO2 conductivity values pertinent to the results depicted in the figure 7.
We express our sincere gratitude to Reviewer 2 for their constructive comments which have been pivotal in refining the quality of our manuscript.